# Innovative Remote Sensing Methodologies for Kenyan Land Tenure Mapping

**Mila Koeva** [1,*] , **Claudia Stöcker** [1] , **Sophie Crommelinck** [1] , **Serene Ho** [2] ,
**Malumbo Chipofya** [3] , **Jan Sahib** [3] , **Rohan Bennett** [4,5] , **Jaap Zevenbergen** [1] ,
**George Vosselman** [1] , **Christiaan Lemmen** [1,5] , **Joep Crompvoets** [2] , **Ine Buntinx** [2] ,
**Gordon Wayumba** [6] , **Robert Wayumba** [6] , **Peter Ochieng Odwe** [6] , **George Ted Osewe** [6] ,
**Beatrice Chika** [6] **and Valerie Pattyn** [7]

1   Faculty of Geo-Information Science and Earth Observation (ITC), University of Twente, 7514 AE Enschede,
    The Netherlands; e.c.stocker@utwente.nl (C.S.); s.crommelinck@utwente.nl (S.C.);
    j.a.zevenbergen@utwente.nl (J.Z.); george.vosselman@utwente.nl (G.V.); Chrit.Lemmen@kadaster.nl (C.L.)
2   Public Governance Institute, 3000 KU Leuven, Belgium; serene.ho@kuleuven.be (S.H.);
    joep.crompvoets@kuleuven.be (J.C.); ine.buntinx@kuleuven.be (I.B.)
3   Westfälische Wilhelms-Universität Münster, Institute for Geoinformatics is part of the Geosciences
    department of the faculty of Mathematics and Natural Sciences, 48149 Münster, Germany;
    mchipofya@uni-muenster.de (M.C.); sahib.jan@uni-muenster.de (J.S.)
4   Swinburne Business School, University of Technology Swinburne, Hawthorn campus BA1231, Australia;
    rohanbennett@swin.edu.au
5   Kadaster International, 7311 KZ Apeldoorn, The Netherlands
6   Technical University of Kenya, Nairobi P.O. BOX 52428–00200, Kenya; gwayumba@tukenya.ac.ke (G.W.);
    rwayumba@tukenya.ac.ke (R.W.); vc@tukenya.ac.ke (P.O.O.); George.Osewe@tukenya.ac.ke (G.T.O.);
    beatrice.chika@tukenya.ac.ke (B.C.)
7   Institute of Public Administration, Faculty Governance and Global Affairs, Leiden University,
    2511 DP The Hague, The Netherlands; v.e.pattyn@fgga.leidenuniv.nl
*   Correspondence: m.n.koeva@utwente.nl; Tel.: +31-53487-4410

**Abstract:** There exists a demand for effective land administration systems that can support the protection of unrecorded land rights, thereby assisting to reduce poverty and support national development—in alignment with target 1.4 of UN Sustainable Development Goals (SDGs). It is estimated that only 30% of the world's population has documented land rights recorded within a formal land administration system. In response, we developed, adapted, applied, and tested innovative remote sensing methodologies to support land rights mapping, including (1) a unique ontological analysis approach using smart sketch maps (SmartSkeMa); (2) unmanned aerial vehicle application (UAV); and (3) automatic boundary extraction (ABE) techniques, based on the acquired UAV images. To assess the applicability of the remote sensing methodologies several aspects were studied: (1) user needs, (2) the proposed methodologies responses to those needs, and (3) examine broader governance implications related to scaling the suggested approaches. The case location of Kajiado, Kenya is selected. A combination of quantitative and qualitative results resulted from fieldwork and workshops, taking into account both social and technical aspects. The results show that SmartSkeMa was potentially a versatile and community-responsive land data acquisition tool requiring little expertise to be used, UAVs were identified as having a high potential for creating up-to-date base maps able to support the current land administration system, and automatic boundary extraction is an effective method to demarcate physical and visible boundaries compared to traditional methodologies and manual delineation for land tenure mapping activities.

**Keywords:** fit-for-purpose; land tenure; land administration; cadastre; UAV; feature extraction; needs assessment

---

## 1. Introduction

The first goal of the sustainable development goals (SDGs)—target 1.4—set by the United Nations (UN) aims to deliver tenure security for all [1]. Strategies to support this goal rely in part on the development of land administration systems (LAS) that formalize land rights that support secure land markets, facilitate poverty reduction and support national development [2]. Broadly speaking, LAS can provide the infrastructure for implementing land-related policies and management strategies and maintain information about people and land involving different organizations, processes, and technologies [3].

Contemporary land administration incorporates the concept of cadastre and land registration, often with a specific focus on the security of land rights [4]. It conceptually fits within the broader land management paradigm [5] with its four land administration functions (land tenure, land value, land use, and land development), ultimately seeking to deliver sustainable development. These functions utilize an underlying land information infrastructure including reliable remote sensing data. It should be noted that in this paper cadastre is considered synonymous with land registry and land administration system.

In sub-Saharan Africa, and in the other developing regions, numerous activities for land tenure recording have been, and continue to be, initiated. For example, in alignment with the SDGs, the Global Land Tool Network (GLTN), an international network of partners setting a global agenda for the improvement of land management and tenure security, develops the so called Social Tenure Domain Model (STDM), a tool for registering formal, informal, group, or individual rights [6].

However, it is estimated that only 30% of the world's population has documented land rights and has access to a formal cadastral system [7,8]. Cadastral mapping is proven as the most expensive part of the land administration system [5]; therefore, there is a clear need for innovation for fast, accurate, and cost-effective land rights mapping. Existing approaches using traditional methods including field surveys often prove to be time-consuming, costly, and labor-intensive.

In response, fit-for-purpose (FFP) land administration suggests technologies should be developed, adapted, selected, and applied to match the capacity and cost constraints of a specific context [4]. The main idea of the FFP approach is to ensure land tenure recording is delivered at scale on a regional and national level, rather than focusing on highly accurate solutions, with less coverage. Three main FFP characteristics are that the land administration systems should focus on the purpose, flexibility and upgradability. The concepts of FFP include principles that cover spatial, legal and institutional aspects on a country level. One of the key principles of the FFP approach is using "general" boundaries extracted by visual interpretation based on aerial images rather than "fixed" boundaries demarcated in the field and measured by a high precision Global Navigation Satellite System (GNSS) technology [9]. Some successful examples where the FFP approach was applied are Rwanda where a Land Tenure Regularisation Program (LTRP) was implemented, and Namibia and Ethiopia with their communal land registration and cadastral mapping [4]. To apply these principles of obtaining general boundaries cheaper and faster, there is a clear need for a new generation of tools and applications that are transparent and scalable [10].

In response, we are developing innovative, scalable methodologies, using remote sensing data and cadastral intelligence, based on fit-for purpose principles to respond to the continuum of land rights [11,12]. The aim of this paper is to assess user needs (in terms of land administration functions), and how the three remote sensing methodologies under development can meet these needs. It also considers how the adoption of these technologies may have governance implications. For the assessment we take into account the above mentioned land management paradigm [5] and FFP spatial

and scalability requirements [4] for a case study located in Kenya. The combination of quantitative and qualitative results collected from fieldwork and workshops, taking into account both social (needs assessment and governance) and technical aspects (developed technologies), makes this paper a significant contribution.

The developed remote sensing methodologies for this study include (1) a unique ontological analysis approach using smart sketch maps, (2) unmanned aerial vehicles (UAV) for mapping procedures, and (3) Automatic Boundary Extraction (ABE), based on the acquired UAV images. The sketch maps mentioned above are hand-drawn either on a blank piece of paper or as annotations over existing spatial information, such as cartographic maps, aerial images, or other maps produced via community mapping. As people draw sketch maps based on observations and not based on measurements, the information is not georeferenced, but qualitative relations of the sketched information can usually be considered as correct (with respect to the background information) [13]. The usage of UAVs as cheap, affordable, and easy to use acquisition technology for obtaining high-resolution imagery is emerging for many applications [14,15]. Their applicability in the domain of land mapping was also explored extensively in a different contexts [16–21]. However, there is a lack of studies that evaluate the appropriateness of UAV technology considering the local context and the fit-for-purpose approach. The high-resolution images are usually used for manual delineation of visual boundaries with additional information attached including the ownership and value of the land [22]. However, manual delineation is time-consuming. To register unrecorded land rights more effective in terms of cost and time, innovative and scalable solutions were explored [23–25]. There are clear advantages in using ABE methodologies, therefore, new tools and techniques were developed for scaling up the mapping procedures in support of indirect cadastral surveying based on remotely sensed data [26,27].

The multidisciplinary nature of the current work, using different integrated approaches, and emerging remote sensing technologies, is novel and innovative to the land administration domain. The paper reports on the findings after fieldworks and workshops organized in Kenya, with the purpose of assessing the needs and end users' readiness, the applicability of the developed remote sensing methodologies, considering the needs and how they may affect the governance aspects. In the background section information related to the previous and current land administration system is explained and the study area is described. The overall methodology of the paper is explained in Section 3 and the concrete methods used for each of the assessed remote sensing methodologies are presented. The results are presented in section four, followed by critical discussion, conclusions, and suggested further steps.

## 2. Case Background and Study Area

Kenyan urbanization to date has been one of imbalanced growth due to ad hoc identification of urban areas, resulting in skewed distribution and inequality in development [28]. This challenge of land governance has been found to be a significant factor in constraining inclusive prosperity more generally across Africa's urbanization phenomenon [29]. It is particularly evident in contested peri-urban lands emerging as a result of metropolitan sprawl across sub-Saharan Africa [30,31]. In response, one of the key strategies consistently advocated by the international development community is the establishment (or improvement) of a formal land market. Such a techno-economic orientation and focus on market-driven urbanization is evident in many contemporary studies of land tenure that continues to pay limited attention to underlying political aspects of tenure regimes [32]. Land tenure is inherently social and political, and in Kenya, land is also overtly cultural. Attention to the cultural aspects of land is particularly relevant in urbanization in Kenya as, first, a majority of Kenya's land resources are held under customary tenure systems, and second, having remained unrecognized by formal systems since colonial rule, indigenous groups have long borne the burden of Kenya's structural adjustments, which have resulted in dispossession and longstanding tenure insecurity [33].

Situated in East Africa, Kenya covers almost 600,000 km$^2$ constituted of 47 counties with a population of almost 45 million [34]. Approximately 80% of Kenya's land is categorized as arid or semi-arid, with only 15% of this suitable—and fully used—for agricultural production [35]. Since 1963, the land administration in Kenya is under the Ministry of Lands, Housing and Urban Development. The organization structure is presented in [36]. Under colonial and post-independence governments, Kenya has operated two land tenure systems simultaneously: statutory (based on English property law) and customary. The 2010 Constitution now recognizes customary tenure systems as Kenya's third type of legal tenure, but administrative implementation of this recognition remains in its infancy. The National Land Commission (NLC) is tasked with oversight for all planning processes in Kenya. In relation to FFP, Kenya is one of the countries that first introduced this approach in 1954. Under this major land reform program, land consolidation and systematic adjudication methods were used to determine the parcel boundaries in the rural parts of Kenya. These boundaries were identified, walked, and demarcated by the local inhabitants, based on aerial images, thus it was a participatory approach. As a result so called Preliminary Index Diagrams (PIDs) were produced, which were used to register the rural land parcels in Kenya for many years [37]. Generally, Kenyan cadaster consists of different types of maps, such as survey plans, field notes, registry index maps, aerial photographs, topo-cadastral maps, deed plans, and title deeds, sometimes with variety of names and accuracy [38]. However, most of them are in paper format and are kept in archives (Figure 1). There has been a research also on integrating the buildings into databased and adapting the existing land administration system according to the international ISO: 19152 Land Administration Domain (LADM) standard [36].

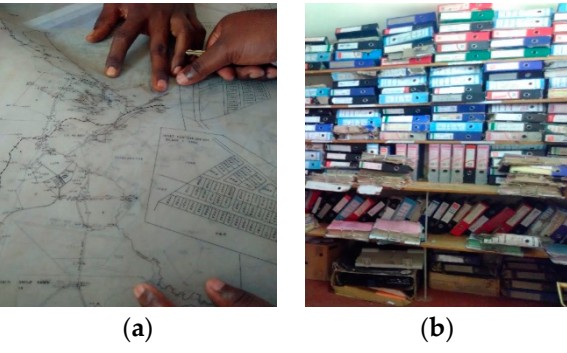

(**a**) (**b**)

**Figure 1.** (**a**) Cadastral map of part of Kajiado; (**b**) archive of land titles.

Kajiado is selected as the case study site for this study as it is part of the Nairobi metropolitan region and is the site of multiple contests for land (Figure 2). Its proximity to Nairobi and Amboseli national parks has also led to increasing human–wildlife conflict being experienced in Kajiado. Currently, approximately 25% of the county's population (of more than 800,000) is urban, almost 50% live below the poverty line, and the population growth rate of 5.5% is higher than the national average [39].

The current land registry map is riddled with information errors stemming from inappropriately scaled maps (resulting in scale errors and boundary overlaps)—the continued use of which introduces further errors in the land registry—which makes it now difficult or impossible to fit new development plans on the original map base [35]. These information issues have resulted in administrative challenges.

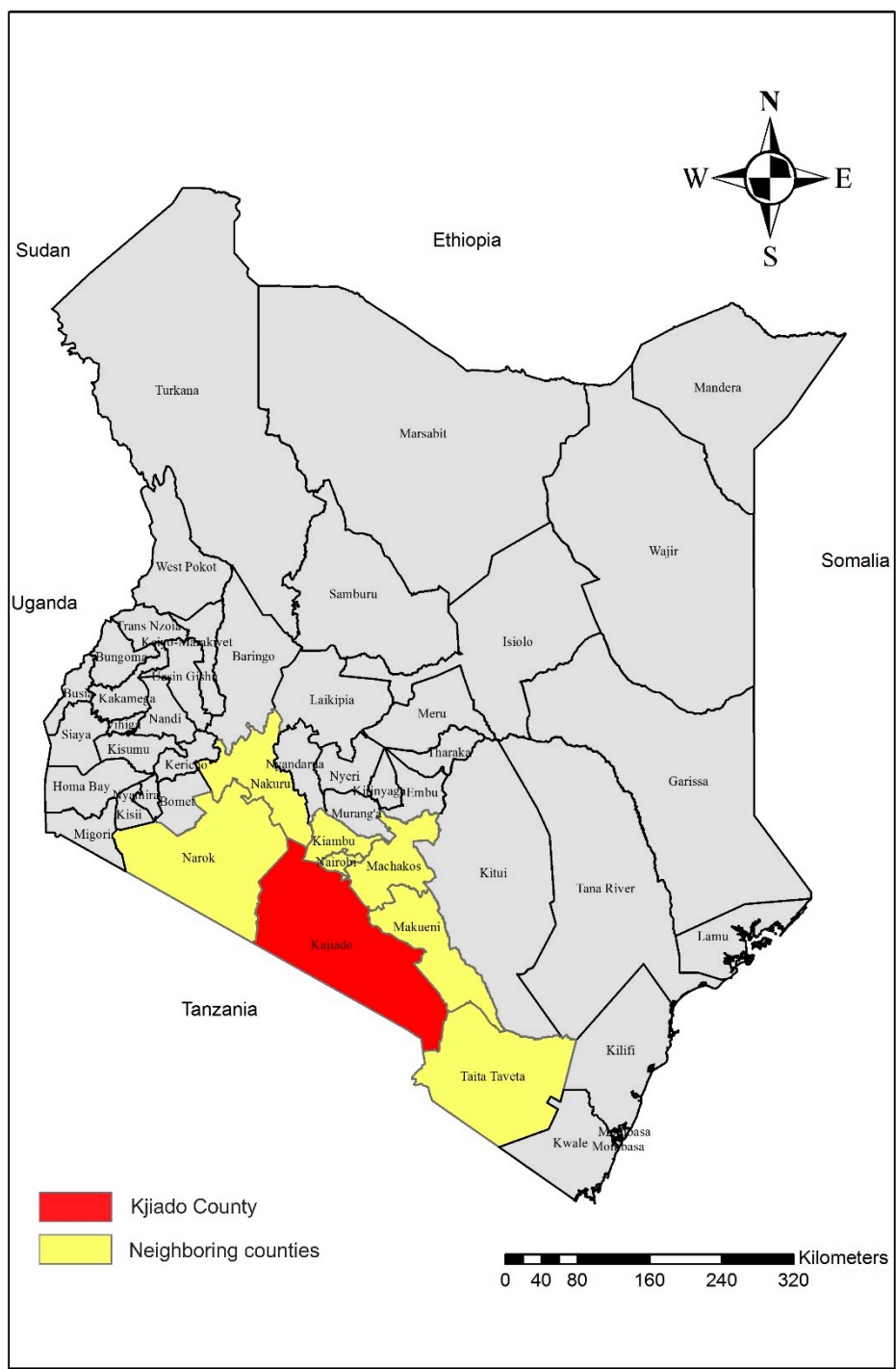

**Figure 2.** Location of Kajiado County and neighboring counties.

## 3. Materials and Methods

Land management activities rely upon three main components [5]: (1) land policy, (2) land administration functions, and (3) land information infrastructure. In the current research, the major focus is the land administration functions including (1) land tenure, (2) land value, (3) land use, and (4) land development. Specifically, this research is mainly seeking to contribute to improving land tenure including cadastral surveys of determining spatial information on parcel boundaries.

First, user needs were identified. Second, the identified needs were incorporated into a broader assessment of the three remote sensing methodologies, made up of 10 criteria, taken selectively from

the land management paradigm [5] and FFP requirements [4]. Third, the governance aspects in relation to the new developments are also analyzed. The entire evaluation methodology is visualized as a conceptual framework in the following Figure 3.

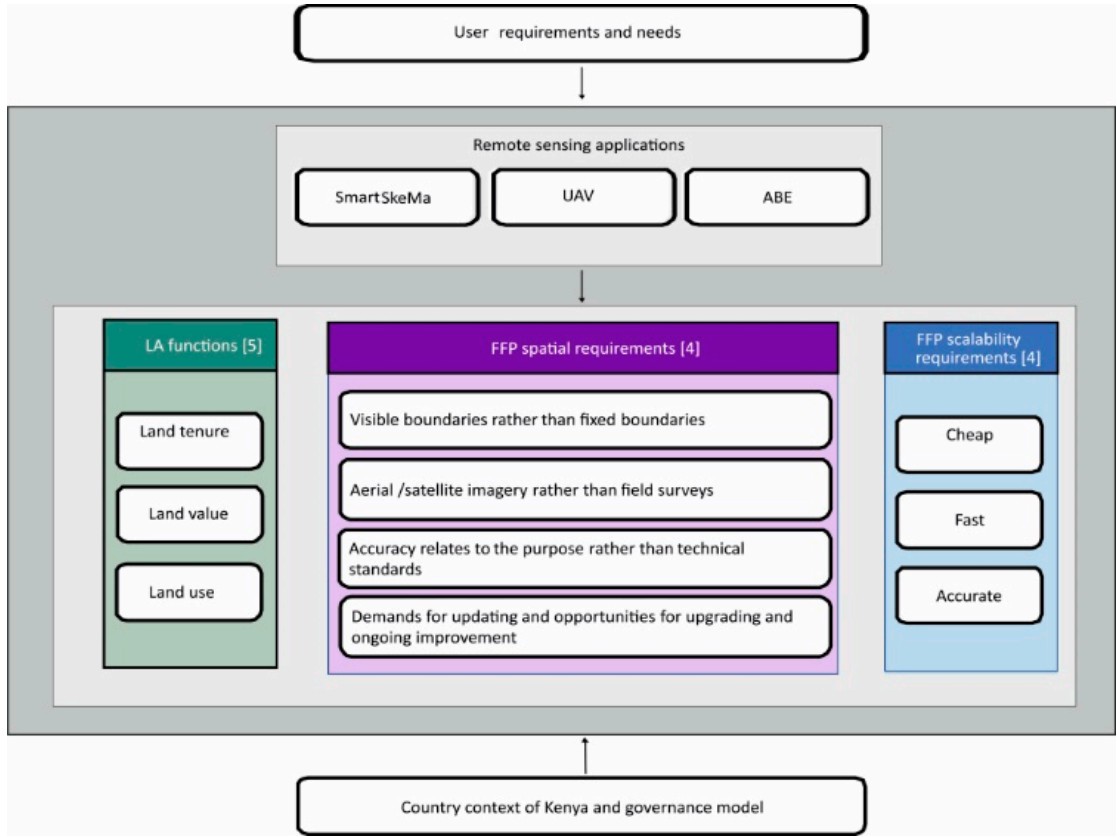

**Figure 3.** Conceptual evaluation framework.

### 3.1. Method for User Needs Assessment

First, for the user needs assessment, assessment was conducted using a behavioral-science-based method called the Nominal Group Technique (NGT). It was developed as a group process model to support the identification and prioritization of problems and/or solutions amongst groups of stakeholders by facilitating equal participation [40,41]. NGT was selected as it draws on individuals' knowledge and expertise while mitigating power dynamics in group-based data collection scenarios [42]. It also produces outcomes that have been found to be robust and meaningful while still being time- and resource-efficient process [43]. A more detailed explanation is provided in [44]. The assessment of the needs was completed prior to other fieldwork: Data was collected from representatives from a range of county government office functions (surveying, registration, planning), as well as county-level officials. Additional data was obtained from local communities with 35 community members from various Maasai families participating (25 men, 10 women).

### 3.2. Remote Sensing Methodologies

Second, fieldworks, workshops, semistructured interviews, and focus group discussions regarding the three remote sensing methodologies were then conducted. This primarily took place in Kajiado from 22th of September to 5th of October 2018. All the workshops were held at the Regional Centre for Mapping Resources in Kajiado. Overall, three one-day workshops with 58 land administration stakeholders from local government institutions, non-governmental organizations (NGOs), private companies, and national government institutions were organized. Each workshop followed the same structure: the project context was presented, participants were split into groups; activities,

demonstrations, and discussions for the SmartSkema, UAVs, ABE, and governance aspects followed as shown on Figure 4 below. A follow-up discussion was held in plenary to produce a strengths, weaknesses, opportunities and threats SWOT analysis of each suggested methodology. Pictures from the workshops assessing the remote sensing methodologies are shown in Figure 4 below.

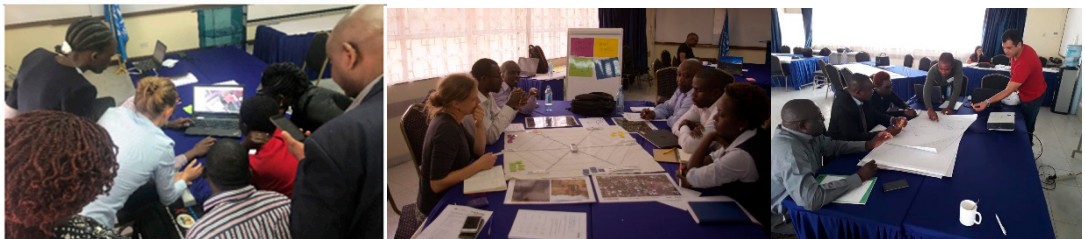

**Figure 4.** Workshops assessing the remote sensing methodologies.

### 3.2.1. SmartSkeMa—Sketch Map Data Collection Software

SmartSkeMa is a software application that we developed to support the documentation of land tenure information for communities with a focus on the actual land practices in the communities. SmartSkeMa supports land recording processes in two main ways [13]. First, it provides a means to document land related concepts as expressed within the local culture or context in a structured domain model [45]. Second, it supports sketch-based community mapping processes by providing a means to digitize, annotate, and geolocalize hand-drawn objects in a sketch map [46]. The method uses both qualitative and quantitative representations of a digitized sketch map and aligns features from the sketch map with corresponding features in the base map. For qualitative representations alignment of qualitative spatial configurations is done. In the case of quantitative (cartesian) representations, the alignment is performed by a coordinate transformation using predetermined control points. The latter approach allows SmartSkeMa to be used as a digitizer over aerial imagery (Figure 5).

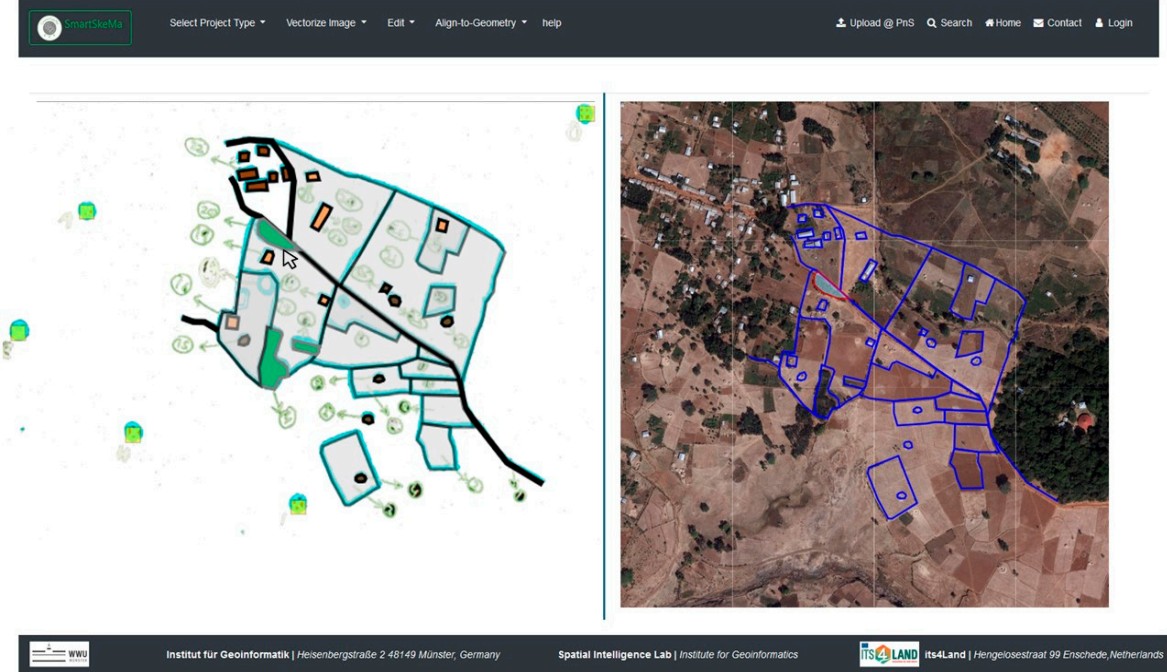

**Figure 5.** Screencast of a SmartSkeMa live demonstration of processing the spatial information drawing on top of a satellite image, the vector representation of drawn features as a SmartSkeMa response, and georeferencing drawn features (web-link: www.smartskema.eu).

Sketch maps are uploaded into SmartSkeMa as raster images. SmartSkeMa then converts these images into vector form in two steps. First any symbols in an image are detected and recognized using a Convolutional Neural Networks (CNN) trained on a set of predefined hand-drawn symbols. The symbols form a visual language for representing land use concepts and land features. After symbol detection the system performs a stroke-based image segmentation wherein boundaries of sketched objects are traced and extracted. Finally, the concepts corresponding to the detected symbols are applied to the image segments based on distance and a fixed set of rules specifying spatial constraints on configurations of different types of features.

The data collection used for the current study needed for the SmartSkeMa system was completed during a series of fieldworks and workshops with male and female members of the Massai community starting from 2017 in Kajiado county and Nairobi, Kenya and running through to October 2018. The sessions included demonstrations of the three main functional parts of the SmartSkeMa (Figure 6), followed by discussions about the applicability of SmartSkeMa. Questions were posed through questionnaires to evaluate the applicability of SmartSkeMa in (1) standard (official) land information recording processes, (2) documenting local land tenure systems, and (3) other land administration tasks.

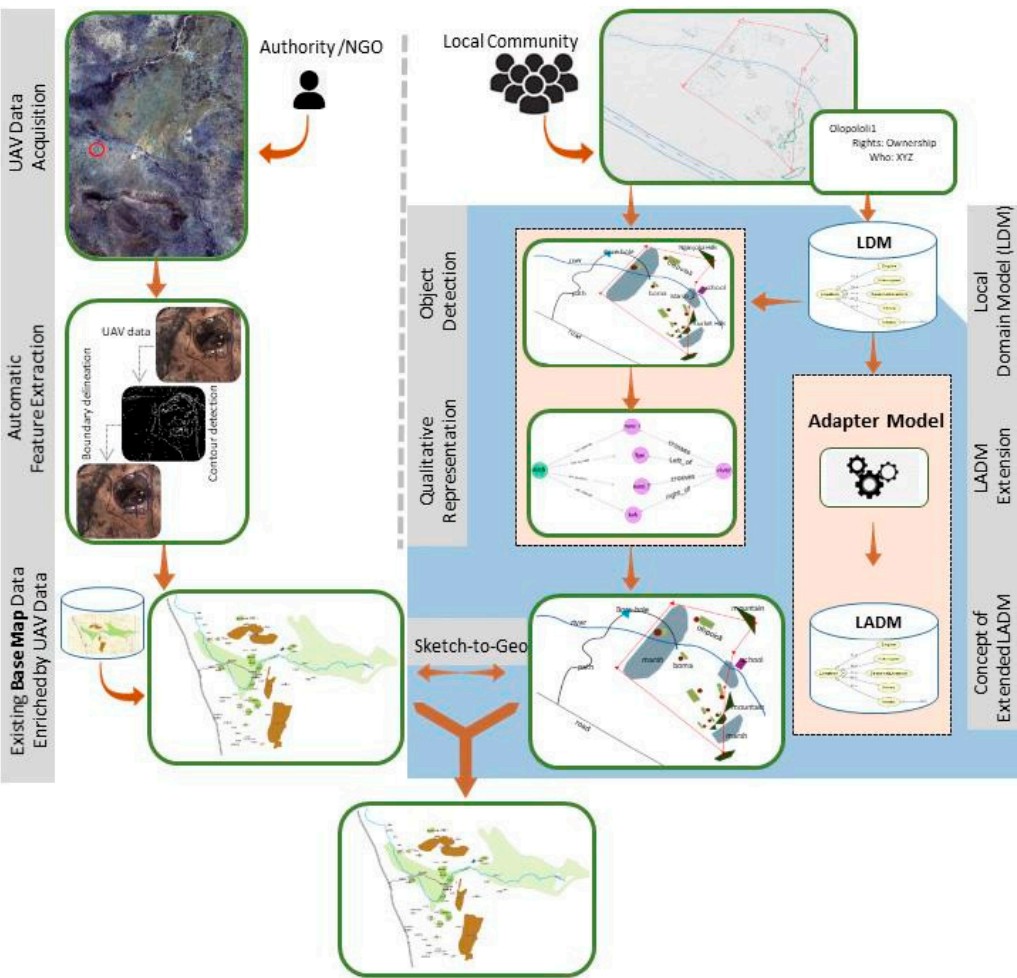

**Figure 6.** Workflow of SmartSkeMa: Right side: local communities provide spatial and nonspatial information via sketch maps. Nonspatial information is processed via local domain model (LDM) and connected via the adapter model to land administration domain model (LADM). Spatial information is recognized via the object detection technique, captured qualitatively via the qualitative representations, and aligned with existing dataset such as feature extracted from UAV data.

### 3.2.2. UAV Data Collection Methods

To prove the concept of UAV data capture as a remote sensing technology for land rights mapping in Kenya, an exploratory research investigation was undertaken. This included the entire UAV-based workflow, starting from the choice of UAV equipment, pilot and flight training, flight authorization, and the final data collection in the field which was carried out in two different sites in Kajiado County: a rural area in Mailua and the township of Kajiado. To accommodate the different characteristics of the flight locations, two different UAVs were chosen (see Figure 7) both with RGB sensors on board.

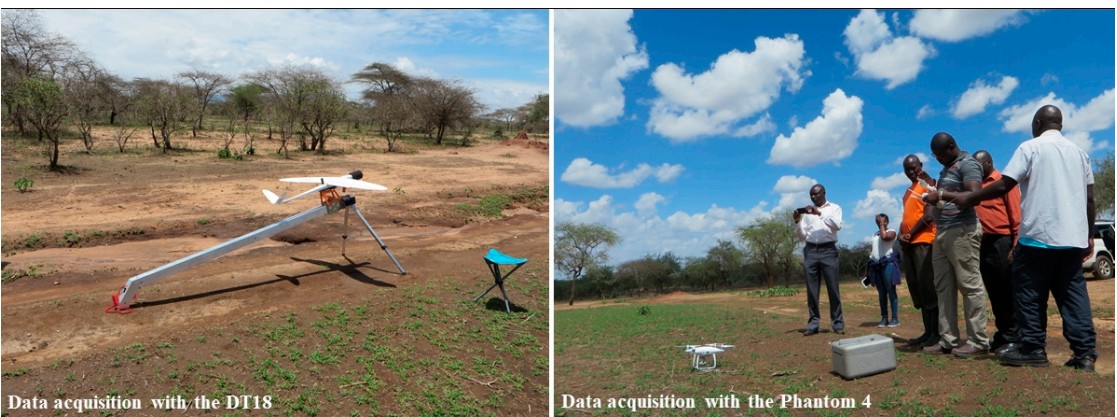

**Figure 7.** UAV data collection with the DT18 in Mailua and the Phantom 4 in Kajiado.

In Mailua, the DT18, a fixed-wing UAV with a long endurance and a large range was selected. In contrast, the vertical take-off and landing UAV DJI Phantom 4 was the preferred equipment to capture data of Kajiado, as the urban area did not provide large spaces for take-off and landing. Both study sites, were captured with indirect georeferencing (Figure 8), i.e., Ground Control Points (GCPs) were distributed within the field and measured with a Global Navigation Satellite System (GNSS) achieving a final accuracy of less than 2 cm. RGB orthomosaics and digital surface models (DSM) of approximately 6 cm Ground Sample Distance (GSD) were generated with Pix4DMapper. Three tiles of 300 × 300 m were selected to demonstrate the boundary mapping approach.

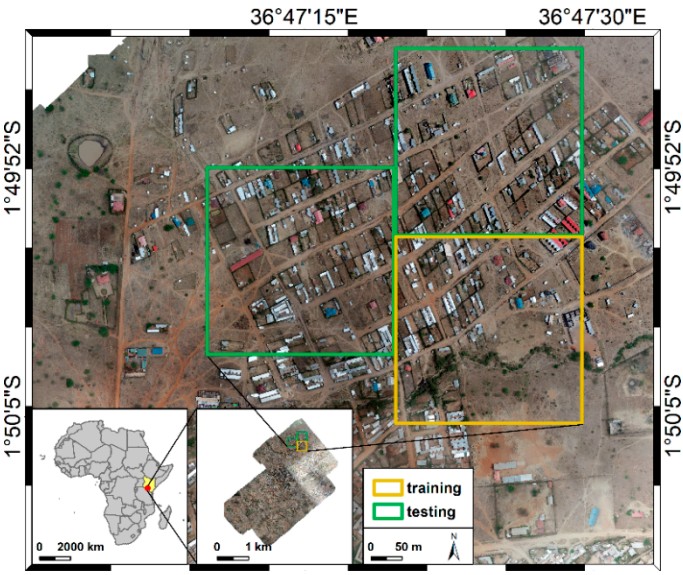

**Figure 8.** Areas of investigation of 300 × 300 m and a 6 cm GSD of Kajiado, Kenya.

The evaluation of the UAV workflow was based on the case study results from Kenya, as well as statistics of the UAV image processing, and resulted in a SWOT analysis. Further insights were gained from a stakeholder assessment of the potential of UAV-based technology to capture land rights in Kenya [47].

### 3.2.3. Automatic Boundary Extraction Methods

The method used for the current study is based on the work in [48] and shown in Figure 9. It supports the delineation of boundaries by automatically retrieving information from RGB data that is then used to guide an interactive delineation. It consists of three parts: (a) image segmentation, (b) boundary classification, and (c) interactive delineation. The source code is publically available [49].

(a) Image segmentation delivers closed contours capturing the outlines of visible objects in the image. Multiresolution combinatorial grouping (MCG) [50] has shown to be applicable on high-resolution UAV data and to deliver accurate closed contours of visible objects [48].

(b) Boundary classification requires labeling the contours from (i) into "boundary" and "not boundary" to generate training data. A set of features is calculated per line capturing its geometry (i.e., length, number of vertices, azimuth, and sinuosity) and its spatial context (i.e., gradients of RGB and DSM underlying the line). These features together with the labels are used to train a Random Forest (RF) classifier [51]. The trained classifier predicts boundary likelihoods for unseen testing data for which the same features have been calculated, as indicated with training and testing. An open-source RF implementation [52] is used.

(c) Interactive delineation allows a user to start the actual delineation process: the RGB orthomosaic is displayed to the user, who is asked to interactively delineate final parcels based on the automatically generated lines and their boundary likelihoods. A user can make use of four functionalities that simplify, vary and speed up the delineation process. We implemented (c) as publically available plugin [49] for the open-source geographic information system QGIS [53].

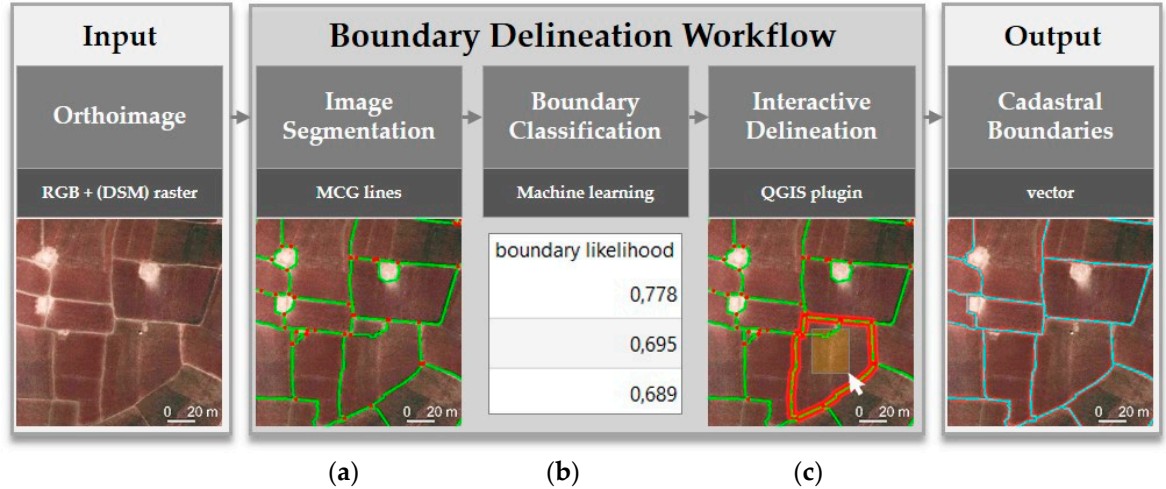

(a)　　　　　　　　　(b)　　　　　　　　　(c)

**Figure 9.** Boundary mapping method: (**a**) Multiresolution combinatorial grouping (MCG) image segmentation. (**b**) boundary classification that requires line labeling into "boundary" and "not boundary" for training. The labeled lines are used together with line-based features to train a Random Forest classifier that generates boundary likelihoods for testing. (**c**) interactive delineation guided by a QGIS plugin.

An analysis on our study area in Kajiado assessed the ABE application for extracting visible cadastral boundaries using the acquired UAV images [24]. During the workshops, we used the three tiles of 300 × 300 m shown in Figure 7 to demonstrate the boundary mapping method. In this

way, we related the method to our conceptual framework (Figure 2). This allowed identifying and understanding bottlenecks: (i) the operational analysis questions when and why the method works better or worse compared to manual delineation and (ii) the feedback analysis investigates the method based on surveying stakeholders responses. The feedback analysis was based on discussing the strengths, weaknesses, opportunities, and threats (SWOT) of our proposed method compared to manual delineation as identified by the workshop participants.

## 3.3. Governance Aspects

To get an overview of the governance requirements to support the adoption and use of the above-mentioned remote sensing methodologies, in-depth semistructured interviews were conducted and focus group discussions were organized. The assessed governance aspects are strongly based on FFP demands [4] and dimensions of the governance assessment tool [54]. Three focus group discussions were organized. The first group focused on local government (with 18 participants), the second on private companies and NGOs (with 32 participants), and the third on national government (with three participants). During the focus groups, participants were able to map the different governance requirements (responsible actors, partners, levels, instruments, resources, organizational characteristics, capacity development characteristics, and cultural characteristics) needed to successfully adopt the technical applications. Besides these focus group discussions, we were also able to do 18 in-depth semistructured interviews in Kenya. For these interviews, we were making use of a guiding topic list to facilitate the extensive data collection to support the development of multi-sectoral profiles (e.g., socioeconomic characteristics, geospatial innovation trends, etc.) of the identified case areas pertaining to land tenure information. The topic list that guided the interviews was a compilation from an extensive literature reviews on governance and capacity development. The questions of the semistructured interviews were not only structured by the specific topics, but also open enough to allow for clarifications, new insights and deepening of the subjects by new, unexpected responses during the interviews.

## 4. Results

### 4.1. Results of Needs Assessment

A range of land administration needs were identified (Table 1) for Kajiado county stakeholders, categorized along the main land administration functions of tenure, value, use, and development (Figure 2); however, other needs (such as governance) also emerged.

**Table 1.** Assessing the needs.

| Land Admin Needs | County Government Needs | Community Needs |
|---|---|---|
| **Tenure** | - Georeferenced land information connected to the registry index map<br>- Land subdivision data<br>- Updated land information<br>- Resurvey of adjudicated areas of public utilities<br>- Good practices related to surveying and mapping | - Updated land subdivision information<br>- Right-of-way information about government wayleaves<br>- Reduce fencing around properties owned by non-Maasai |
| **Value** | - No of properties (and its attributes) in the county | - Locate and protect culturally significant resources (e.g., important waterways)<br>- Map areas of cultural value and culturally significant objects (e.g., trees). |

**Table 1.** *Cont.*

| Land Admin Needs | County Government Needs | Community Needs |
|---|---|---|
| **Use** | - Approval and placement of private boreholes<br>- Environmental degradation<br>- Documentation about the location of public utilities<br>- Information about cottage industries (as these are damaging properties) | - Understand and respect Maasai land use practices<br>- Improve poor animal husbandry practices<br>- Manage overgrazing<br>- Improve wildlife corridors to reduce loss of land<br>- Information about wild animal infestations needed due to damage infrastructure and spread diseases |
| **Development** | - County spatial plan<br>- Land use zoning and development controls need to be defined<br>- Land fragmentation (subdivisions too small)<br>- Map and demarcate roads to avoid informal development<br>- Clearly mark ecologically fragile areas | - Define migratory corridors and restrict sale or give right-of-way to Maasai<br>- Identify and document fertile grazing areas at waterways that need to be mapped and preserved<br>- Improve drought mitigation<br>- Mitigate deforestation |
| **Governance** | - Relationship between land laws<br>- National vs. county land policy<br>- Improving data management for multipurpose use<br>- Community understanding of women's rights in land transactions<br>- Improve understanding of community's land needs and improve engagement around land policy | - Need to integrate community knowledge with formal and/or statutory information.<br>- Legal aspects of land conflict not well understood by community |

Land tenure security is identified as a fundamental need in Kajiado, and is being challenged by urbanization. In interviews, the county estimated that almost 80% of its registry's resources were directed towards resolving land disputes (e.g., in ground truthing and reporting). Unsurprisingly, the needs assessment reflects this: land tenure needs identified included improving the quality of registry information, especially spatial information; associated with this is the need to improve subdivision data in general—both in terms of data and processes—and more generally, the need for updated information. Other tenure needs related to spatial and administrative information pertaining to public utilities. Similarly, communities identified the need for better information about subdivisions (especially within group ranches) and the spatial extent of government wayleaves and associated rights. This was important to support understanding how land is acquired to protect right-of-way for maintaining public infrastructure and compensation, provided around reduced use rights.

The information needs identified around land value, use and development were not as easy to differentiate: all three functions of land are interlinked. From the government's perspective, identified needs with direct implications for land value were around better-quality information about the number of properties in the county. For the community, the identified needs reflected the need for preservation of culturally significant areas (e.g., important waterways) and objects (e.g., trees).

There were greater needs identified around land use, and these related to lack of knowledge around where boreholes were being placed and used (which is draining the local water table), the increasing

impact of drought, and mitigating general environmental degradation, especially resulting from unregulated cottage industries, such as charcoal production (where burning of trees also impacts the value of private properties). Similarly, for communities, needs identified were around mitigating unsustainable land use practices that were either impacting the Maasai way of life, or draining environmental resources that impacted on their ability to rear livestock. While not directly referencing land information, most of the needs certainly infer some type of spatially-enabled decision-making, e.g., knowing where overgrazing occurs, and knowing where existing wildlife corridors are, or where to situate new ones.

The broader issues implicated in the identified land use needs are also reflected in the land development needs. Kajiado is rapidly urbanizing; consequently, the county government would like to better understand how to plan and manage development. This included the need for better spatial planning (through production of a county spatial plan), better planning controls (through zoning) (especially as land fragmentation is becoming an issue), and defining ecologically fragile areas. For the Maasai, land information needs around migratory corridors (e.g., restrict sale or give right-of-way encumbrances in favor of Maasai) and fertile waterways reinforce their desire for land use practices that enable them to flourish culturally. Considering the rapid influx of "outsiders", i.e., non-Maasai, into Kajiado, the community emphasized the importance of understanding and respecting Maasai communal-based practices of resource sharing and the implications this will have on property boundaries. However, there are also needs around broader environmental issues caused by over and unmanaged development, such as drought and deforestation, and wild animal infestations which damage property (e.g., water pipes) and spread disease amongst herds.

Finally, the whole range of governance needs emerged, which reflected the disconnect in land information and land policies at national and county levels and the disconnect between government and communities (despite the Constitution enshrining participatory action in land development) around rights (e.g., women's land rights) and responsibilities (e.g., improving community engagement). For the Maasai, additional elements reflected the disconnect between formal and customary knowledge systems (and relevant data), but reinforced the fact that communities do not have a good understanding of legal and policy frameworks pertaining to land (e.g., land conflict), which leaves them vulnerable to poor decision-making.

*4.2. Results from SmartSkeMa*

Stakeholder impressions of the SmartSkeMa application were sought along three main dimensions: (1) ability to support conventional land tenure recording activities, (2) ability to facilitate community driven land tenure recording systems, and (3) applicability in other land administration functions. SmartSkeMa was generally judged to have the necessary functionality to support standard land tenure recording activities. Among 21 participants, 16 considered SmartSkeMa to be usable together with standard land administration systems, while two considered this to not be the case and three were ambivalent (they neither agreed nor disagreed to the statement). In addition, of the 21 participants, 18 agreed that the functionality of SmartSkeMa is useful for recording land tenure information while three mentioned that it was only partly useful for that purpose. The participants also indicated the reasons for their judgements or choice. Table 2 shows a summary of these data coded into themes as presented by the participants.

**Table 2.** Summary of participants perceptions in the usefulness of SmartSkeMa for land tenure recording.

| Usefulness for Land Tenure Recording | Reasons | Comments |
| --- | --- | --- |
| Partly useful | Poor geometric accuracy or poor precision | Poor accuracy or precision will lead to legal impediments. May not work in densely populated areas. |
| Very useful | Can be used to delimit communal land rights; physical planning; updating official maps; delimiting communal land rights; consultation and public participation; reach consensus when recording land rights; record information from community perspective. | Requires interoperability with government systems, government buy-in, and may face legal impediments. |

In terms of facilitating community driven land tenure recording systems SmartSkeMa was considered more favorably. Of the 21 participants, 18 believed that SmartSkeMa could support communities to register and govern their lands according to local customs. There was no clear agreement on which other land administration tasks the SmartSkeMa application could be applied to. Several tasks stood out with land use documentation and land use planning mentioned by six participants; recording of historical and inaccessible information was mentioned by four participants; and aiding surveying and other traditional land information collection was mentioned by three participants. Finally, we asked the participants to perform a SWOT analysis of the tool based only on the functionalities that have been presented to them during the demonstration. The results of this analysis are shown in Table 3 below.

**Table 3.** SWOT results on SmartSkeMa.

| Strengths | Weaknesses |
| --- | --- |
| - Has multiple applications (incl. collecting historical information; creating land use plans; documenting land rights)<br>- Has low barrier to entry<br>- Is participatory<br>- May create trust among community participants<br>- Can help reduce conflicts after parceling<br>- Can produce preliminary data for land surveys | - Time-consuming in the field and during preparation<br>- Cannot yet associate attributes to boundaries<br>- Difficult to get community engaged<br>- Requires background knowledge<br>- Collected data may not be accepted as meeting legal standards for land adjudication due to poor accuracy |
| **Opportunities** | **Threats** |
| - Use in implementation of the Community Land Act's community land registers [55].<br>- Incorporation of Satellite imagery to prepare the base map data and sketches | - Difficult to get community engaged<br>- Misalignment with official records or existing systems |

The feedback obtained from the workshops laid the foundation for the development of the second method: use an aerial image as the background for a sketching exercise. This is expected to increase the precision and provide measurable accuracy. The alignment of a sketch traced on top of an aerial image is done by a 6-parameter affine transformation. The parameters for the transformation are estimated by ordinary least squares linear regression quadratic features.

The new method was tested on a small sample of parcels and three metrics were taken as shown in Table 4. The time for delineation cannot be compared with traditional method since the time to produce the parcels is mostly consumed by the field work. As field work is required to collect parcel information in other approaches as well, we conclude that the automatic delineation of sketch maps results in a faster process.

**Table 4.** Performance metrics of parcel delineation using SmartSkeMa's sketch-on-map.

| Manual Delineation | | |
|---|---|---|
| Number of parcels in the sample | Sketching time per parcel | Mean Deviation from cadastral boundaries in meters (sampled) |
| 9 | 6 min | 1.29 |

### 4.3. Results from UAVs

The case study revealed many opportunities but also a number of challenges for UAV data capture as a technical solution to provide a spatial database for capturing land rights and cadastral boundaries in Kenya. In most countries, before commencing a UAV flight mission, regulatory clearance has to be in place to ensure the safety of airspace users, people, and property on the ground [56]. In that regard, Kenyan UAV legislation underwent changes during the case study. Before the official regulations were gazetted [57], the use of UAVs was heavily restricted, with a mandate to seek flight permission from Ministry of Defense and Kenya Civil Aviation Authority. At the time the regulations were passed, processes for flight authorizations seemed to be straight forward. However, a reality of a too costly and restrictive procedure largely impeded the rise of UAV technology in Kenya. Soon after release in June 2018, the regulations were nullified by the Government, leaving a regulatory vacuum in the country. Both data acquisition flights were carried out with a temporal flight authorization and awareness of the local government.

After an extensive sensitization of the local government and community, the UAV data, as well as GNSS measurements, were completed in March 2018 (Mailua) and September 2018 (Kajiado). The RGB pictures were processed with Pix4D to create an orthophoto (Figure 10). Flight specifications and information on geometric accuracy are summarized in Table 5.

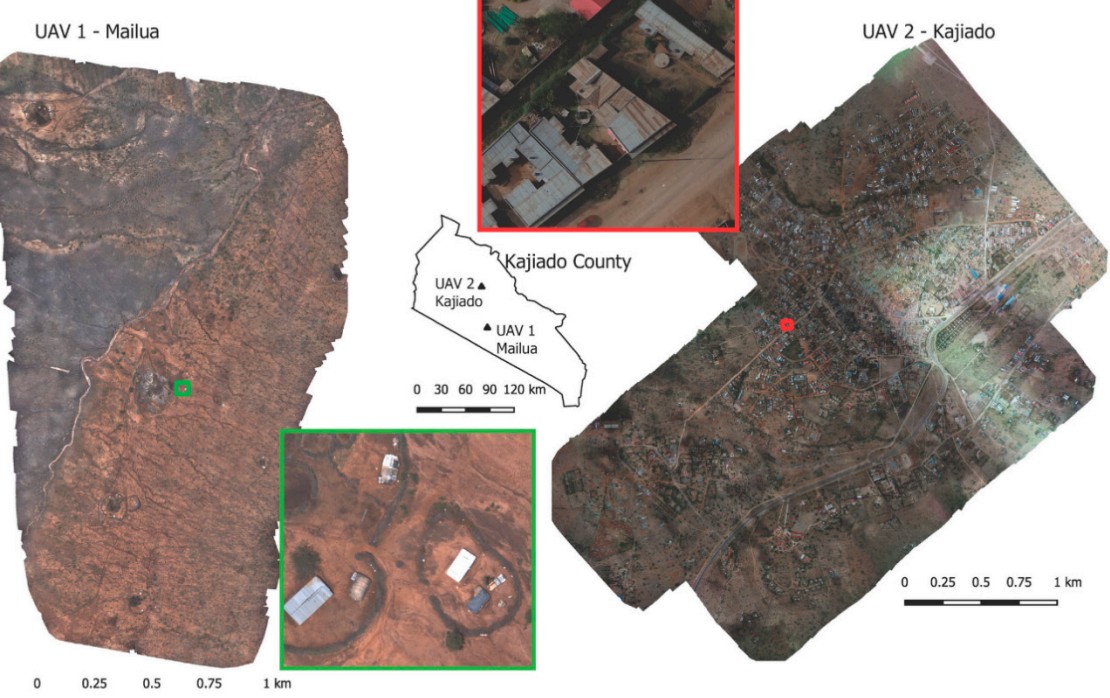

**Figure 10.** Overview of UAV datasets.

**Table 5.** Flight characteristics and geometric accuracy of Kajiado and Mailua dataset.

| | **Mailua** | **Kajiado** |
|---|---|---|
| **UAV equipment/sensor** | DT18 PPK/ DT18 3bands | DJI Phantom 4/inbuilt sensor |
| **UAV Type** | Fixed-wing | Fixed-wing |
| **UAV Sensor [mm]** | $8.45 \times 7.07$ | $8.45 \times 7.07$ |
| **Resolution** | 5 MP | 5 MP |
| **Flight time** | 4 flights á 45 min | 15 flights á 10–15 min |
| **Area captured** | 3.32 km$^2$ | 8.28 km$^2$ |
| **Flight height** | 200 m | 200 m |
| **Overlap forward/side [%]** | 80/70 | 80/70 |
| **Ground Sampling Distance** | 5.72 cm | 5.8 cm |
| **Ground Control Points (GCPs)/** | 8 | 16 |
| **RMS Error (X/Y)** | 0.022 cm/0.024 cm | 0.040 cm/0.042 cm |

However, our case study showed that UAV workflows are easy to transfer to different contexts: data acquisition always follows a standard procedure following the steps of flight planning, data collection and postprocessing. Prices of UAV equipment vary largely, offering technical platforms for almost every budget without compromising too much on data quality. Nevertheless, the purchasing costs might give an indication of the longevity and the reliability of the UAV components, which is beyond the results that the case study currently provides. Similar to the price for UAVs, the accuracy of the final orthomosaic can differ from several centimeters to meters, as it depends on the GNSS sensor of the UAV, the availability of a geodetic network, the visibility of satellites during data acquisition, and the strategy of ground control measurement. The insights from the workshop can be concluded in a SWOT analysis (Table 6).

**Table 6.** SWOT results on UAV data acquisition.

| **Strengths** | **Weaknesses** |
|---|---|
| - Provides reliable data products (orthophoto, 3D point cloud, and digital surface model) for multiple purposes in land administration<br>- Various UAV platforms and sensors can be utilized depending on the context and geographical conditions<br>- Immediate data collection to gather up-to-date base data, if flight permission is granted by the authority<br>- Automated flight planning and image processing reduces training effort<br>- High spatial resolution and geometric accuracy | - Dependent on weather conditions<br>- Limited to small to medium scales<br>- Real-Time Kinematic (RTK) or Post-Processed Kinematic (PPK) workflows require professional GNSS equipment for static observations<br>- Time-consuming measurement of Ground Control Points to reach a high level of geometric accuracy if RTK or PPK workflow is not supported by the UAV equipment |
| **Opportunities** | **Threats** |
| - Community engagement as data is being collected directly in the field<br>- Ease of use allows capacity development at the local level (e.g., for bottom-up initiatives)<br>- High-resolution orthorectified images for cadastral mapping in urban contexts<br>- Low investment costs for decent UAV equipment | - The unclear legal situation that potentially prohibits or restricts UAV flights<br>- Maintenance services of UAV equipment not available in the country<br>- UAV technology not included in current surveying act—a high barrier to adopt the technology |

*4.4. Results from ABE*

Delineating boundaries with indirect surveying from the remote sensing imagery requires knowledge about the boundaries. To recognize boundaries in an image, it helps to be familiar with

their appearance on the ground. We, therefore, went to the area for which UAV data was captured and took images of example boundaries. A team of village elders and a local researcher joined us to communicate with land owners when passing and capturing their boundaries. The team explained which objects were typically used to demarcate boundaries and provided insights on local boundary demarcation challenges.

During fieldwork in Kajiado, we obtained an understanding of local boundary characteristics and demarcation challenges. The letters used in the following refer to Figure 11.

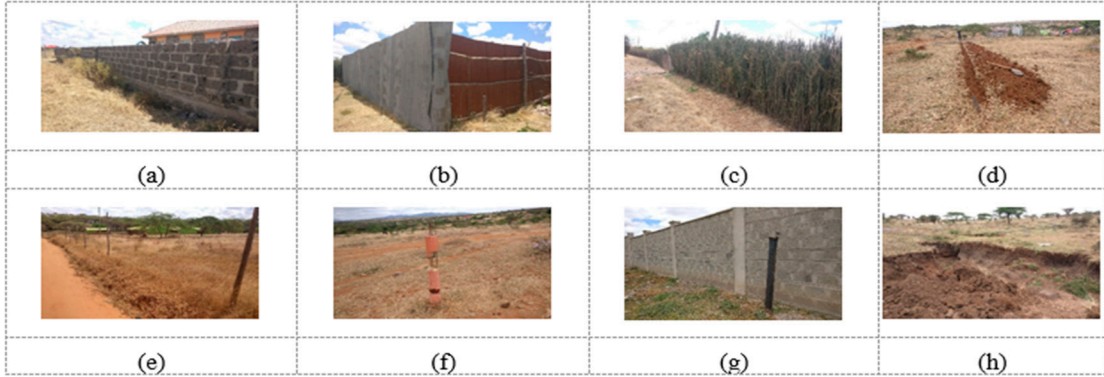

**Figure 11.** (**a**–**d**) Examples of visible boundaries in Kajiado. (**e**–**h**) Boundary demarcations challenging to identify correctly from remote sensing imagery collected during the field survey.

A majority of boundaries are demarcated by visible objects such as (a) stone walls, (b) corrugated metal fences, (c) vegetation, or (d) ditches. The following examples are extractable from remote sensing imagery though require local knowledge or context for a correct identification: (h) ditches can be confused with soil erosion when extracted from imagery alone. (d) Some fences demarcating boundaries are challenging to differentiate from its surrounding. High-resolution digital surface models (DSMs) can support the identification of such fences. (f) Beacons demarcate boundary corner points and (g) can be used in parallel with linear boundary demarcations, or as control points for hosting measurements.

The cadastral boundary has often remained on the connection of the beacons, instead of on the visible boundary. Based on the local knowledge obtained during fieldwork, and the large portion of cadastral boundaries in Kajiado being visible following the FFP principles, the boundary mapping approach could be applied to the captured UAV data (Figure 12).

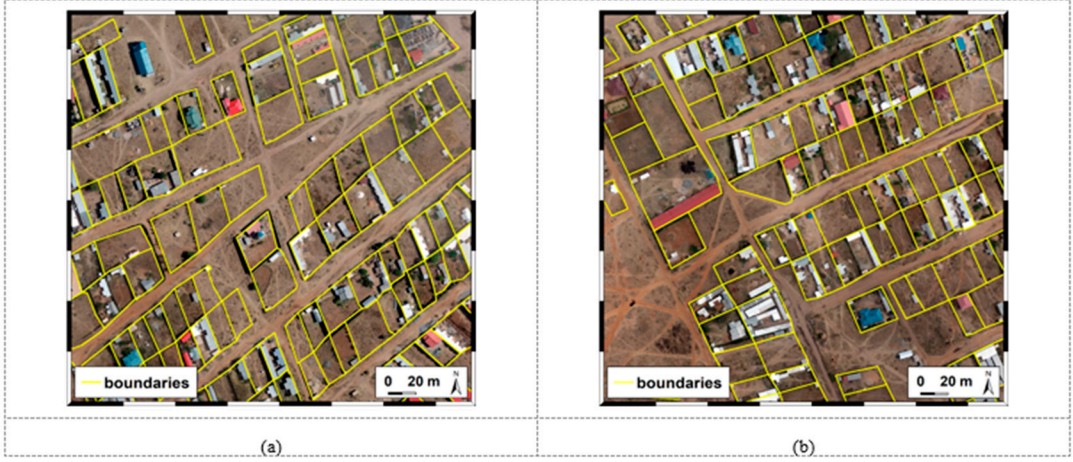

**Figure 12.** (**a**,**b**) Cadastral boundaries delineated from UAV data.

Some challenges that we observed during delineation are shown on Figure 13 below.

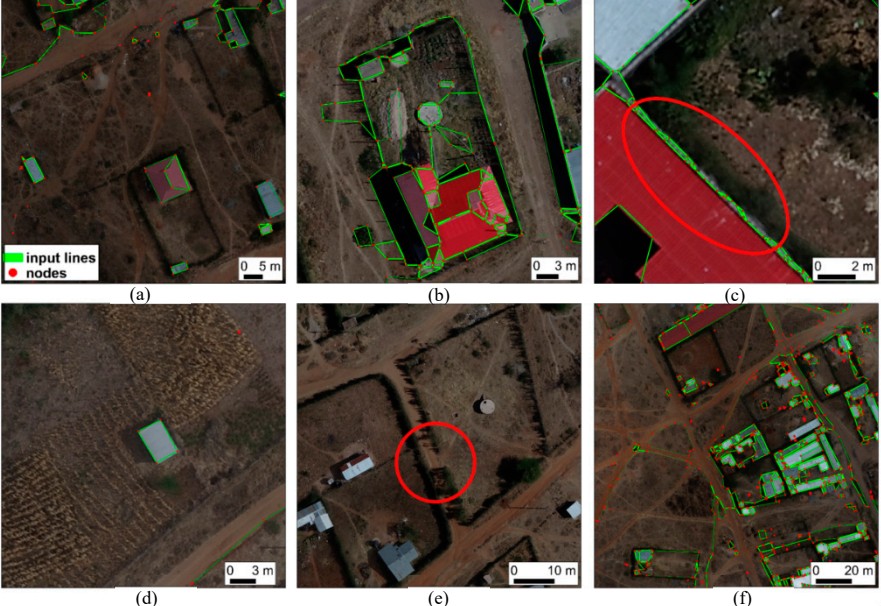

**Figure 13.** Challenges observed during delineation: (**a**) undersegmentation, (**b**) oversegmentation, (**c**) fragmented segmentation, (**d**) redundancy of least-cost-path calculation, (**e**) visible boundary not demarcated by objects, but by context, and (**f**) identification of delineation areas through boundary mapping approach.

Existing reference maps for our area would mostly consist of Registry Index Maps (RIMs). RIMs show the outline of land parcels within a given jurisdiction using general boundaries along visible features. The boundaries' position is only indicative and not legally binding. RIMs and survey plans for urban areas have the highest accuracy specifications of 30 cm nominal positional accuracy [38]. Different types of RIMs exist that partly allow positional errors of up to 200 cm [58,59]. As the digital cadaster coverage in country is low the local experts shared that even meter level accuracy can be acceptable for certain areas. However, we observed how time-consuming and tiring this procedure is. The expert should zoom in and out continuously in searching for visible boundaries. Then, the accuracy for delineation of each parcel will be different depending on the skills and precision of the operator. The automatic approach that was proposed speeded this process. The automatically detected and suggested boundaries just have to be checked by the operator and with several clicks to be adjusted and approved. It was observed that for long curved objects manual delineations is much slower and requires continuous clicking and the automatic one requires to click only on the starting and ending point. For a small rectangular object it is required to click only inside of the object and the boundary will be automatically delineated. Using the new proposed method, we reduced the number of clicks with 80%, saved 38% of the time and achieved 71% accuracy compared to manual delineated boundaries [48].

The operational analysis showed that the approach is most suited for the delineation of visible cadastral boundaries demarcated through physical objects. In our study side area, walls and fences were partly covered by vegetation and not built consistently. From 211 parcels, 21 could be delineated without further editing, 24 required minor editing on <20% of the outline length, and the remaining parcels were digitized through snapping to the automatically generated lines and generating new ones. In general, the approach obtains the highest time savings for areas in which boundaries are visible, long and curved, whereas boundaries in our study side are often covered, short, and straight.

The feedback analysis investigated the strengths, weaknesses, opportunities, and threats (SWOT) of the approach. Feedback is derived from three one-day workshops for 57 land administration stakeholders from local government institutions, NGOs, private companies, and national government institutions. The SWOT feedback from the three workshops is shown on Table 7.

**Table 7.** SWOT results on automated boundary extraction approach.

| Strengths | Weaknesses |
|---|---|
| - Image-based delineation facilitates participatory mapping<br>- Allows accurate delineation of georeferenced boundaries without fieldwork<br>- Adaptable per area and its characteristics<br>- Fast data processing and clear visualization<br>- Intuitive usability<br>- Less zooming and clicking<br>- Open-source implementation<br>- Less monotonous delineation work<br>- Easy to implement for existing workforce with surveying background<br>- Strong match with Kenyan land challenges (subdivision, digitization, transformation from general to fixed boundaries, correction of overlapping boundaries, mapping of unrecorded boundaries) | - Dependence on visible boundaries<br>- Time-consuming for large areas<br>- Dependence on knowledge, interpretation and skills of delineator<br>- Superiority over manual delineation depends on image segmentation and its match with cadastral boundaries<br>- Varying data quality due to lack of standardized image capture/processing<br>- Interactive design limits reproducibility<br>- High initial costs (set up of digital infrastructure, capacity development)<br>- Image acquisition requires equipment, training and permissions<br>- Open-source solution requires acceptance<br>- Update of Registry Index Maps (RIM) not included |
| Opportunities | Threats |
| - Potential to increase superiority over manual delineation by adding functionalities (geometric checks for output lines, creation of polygons/attributes and change protocols)<br>- Modular workflow can be updated in case of future innovations (e.g., on image segmentation or classification)<br>- Applicable in further object delineation applications (e.g., land use mapping)<br>- Implementable in existing systems due to modular design<br>- Potential to reduce land-related disputes through clear visualization and identification of boundaries | - Possible inability to cope with rapid technology changes despite modular design<br>- Superiority over manual delineation too small (reduced efficiency when object outlines do not match with cadastral boundaries, high percentage of invisible boundaries, or beacon demarcation)<br>- Possible non-acceptance of open-source solution (no guarantee for long-term source-code maintenance) and threat from commercial solutions<br>- Resistance to innovative approaches (fear of job loss due to automation)<br>- Uncertain legal allowance to capture/use aerial imagery<br>- Unstable Kenyan digital infrastructure |

The methodology was tested and works also for remote sensing data with different resolutions (0.02–0.25 m) acquired from other platforms such as satellite and aerial cameras on board of an airplane. Advantages are strongest when delineating in rural areas due to the continuous visibility of monotonic boundaries. Manual delineation remains superior in cases where the boundary is not fully visible, i.e., covered by shadow or vegetation. Although our methodology has been developed for cadastral mapping, it can also be used to delineate objects in other application fields, such as land use mapping, agricultural monitoring, topographical mapping, road tracking, or building extraction.

*4.5. Results from Analysis of Governance Aspects*

In the focus group discussions and individual semi-structured interviews held around each remote sensing application, several governance aspects were raised. As most of these apply to two or all three remote sensing methodologies, and are discussed here jointly along the lines of six aspects derived from the discussions: (1) legal versus informal rights, (2) government versus non-governmental actors, (3) the national versus regional/local government, (4) digital versus paper way of working, (5) use of open source software, and (6) lack of clear legislation for specific new tools and applications esp. UAVs.

Many different definitions of the term "governance" exist, but most important is that it stands for a broader concept than government, and also includes the influence of other actors on processes that

affect all. Within the context of the research, a definition was developed where governance is "the process of interactively steering the land tenure society to sustain the use of the its4land tools" [60].

In Kenya the 2010 Constitution brought a number of changes that affect the governance aspects of our remote sensing methodologies. As mentioned earlier, customary tenures are now explicitly recognized in the Constitution, although the attention to them in specific laws and regulations is still lagging, and in peri-urban (and informal) areas, other forms of non-statutory tenure rights exist that are not specifically mentioned. The formal systems for land administration, that tend to only serve statutory rights, are embedded in laws and regulations, but also in the way the different formal land sector actors operate in practice; which tends not to focus on innovation or broadening of the beneficiary group. There is, currently, a lack of participatory mechanisms that can support the collaboration between the different governmental levels and the non-governmental actors. Political interests or corrupt practices were mentioned during the workshops and interviews. These practices happen due to both the lack of transparency in the decision making process and lack of an enforcing institutional environment. Further, there is no specific legislative framework that supports innovative approaches as the ones offered via our developed applications.

Allowing non-governmental organizations (such as private companies, NGOs and professional network associations) to take the lead in implementing the more participatory and innovative technical applications is also difficult. There is not really a tradition to do so, which is partly due to lack of resources: financial, human and technological. Further, the fear of losing jobs due to introduction of new ways of work make the street level bureaucrats wary, whereas the higher level workers fear of loss of the control of the currently used methods which involve political interests and corruption practices. As most of the national government and counties lack basic infrastructure, one way the national government could support the implementation of technical applications is by providing financial or legal incentives to non-governmental actors, as in many cases there are consultancies who have the expertise and could support the adoption of technical applications within a short time frame. However, neither governmental actors nor private companies are used to this type of participatory approaches. Until now, according to the different actors who participated in the workshops, there have not been real participatory approaches that could support their implementation. The capacity of the local levels to implement technical applications like ABE and SmartSkeMa face the challenge of variety in capacity among the counties, and some cases were reported where governmental employees need to use their personal computers to carry out their daily job activities.

The 2010 Constitution brought the devolution of powers to the 47 counties. There is still lack of clarity relating to the division of responsibilities between the county and national government level, and the different governmental levels currently often lack resources to implement, maintain or upgrade the use of innovative technical applications, especially when those require the specific IT knowledge that comes with geospatial techniques. The current governance structure favors a top-down implementation process where the national government is the main actor. While some counties have the capacity to support the implementation of the technical applications, others clearly lack infrastructure, financial resources or knowledge.

In addition to the limited capacity, it also became clear that only some governmental actors see the transition from paper-based data to digital based data as a priority. The transition from paper-based data to digital data is already set in some counties, but is not always perceived as a priority by all governmental actors. Due to the lack of political will, the implementation of our technical applications cannot be expected to occur in the short-term. Political interests or corruption practices around the possible implementation of the technical applications were also mentioned by the different interviewed actors. This situation is due to the lack of legislation for digital data and the current prioritization of paper-based data.

## 5. Discussion

This paper was designed to assess user needs (in terms of land administration functions), how the three remote sensing methodologies under development meet these needs, and finally what governance aspects would be critical in widespread update. As a result of the workshops in Kenya, a SWOT analysis was created for each developed application. The results of those SWOT analyses as well as from the fieldwork are summarized and visualized in Table 8: the adherence to 10 aspects of the assessment criteria, derived from the user needs assessment, land management paradigm [5], and FFP requirements [4] is shown.

**Table 8.** Assessment of remote sensing methodologies with regard to fit-for-purpose land rights mapping in Kenya. Green indicates compliance with an aspect, yellow indicates that the application partially complies with an aspect.

| | Sketch Maps (SmartSkeMa) | UAV-Based Data Collection | Automated Boundary Delineation |
|---|---|---|---|
| 1. Land Tenure | | | |
| 2. Land Value | | | |
| 3. Land Use | | | |
| 4. Visible boundaries rather than fixed boundaries | | | |
| 5. Aerial imagery rather than field surveys | | | |
| 6. Accuracy relates to the purpose | | | |
| 7. Updating and ongoing improvement | | | |
| 8. Cheap | | | |
| 9. Fast | | | |
| 10. Accurate | | | |

With regards to SmartSkeMa, is seems clear that this is not a methodology aiming to replace data collection via aerial images or other surveying techniques, but sketch maps can be used to complement and support collecting data about the relationship of people with respect to land. When SmartSkeMa is considered as methodology for documenting community land tenure in Kenya, its ease of use makes it a cheap option as, once set up, it allows communities to document their land with little additional cost. Its level of accuracy can also be tailored to the task at hand since communities can sketch on top of an aerial image allowing higher precision than is obtained using a plain sketch map. Finally, because a community can use SmartSkeMa with relative independence it may produce data faster than would be possible using traditional land survey methods where the skilled personnel in Kenya are scarce. From the results obtained, SmartSkeMa's functionalities contribute to meeting most of the 10 aspects. The wide range of spatial precision covered by SmartSkeMa presents a great opportunity for incremental and progressive land data acquisition. However, data produced by SmartSkeMa is not very well suited for land valuation in the sense of calculating objective monetary equivalents. The data however may include information about relative values as perceived by land users within a cultural context. More work is needed to determine the extent to which these land values can be captured in the data and how they can be interpreted.

For UAVs, during the workshop most interest was conveyed in the provision of an up-to-date map. Various local government entities such as the department of urban planning and spatial development identified the potential of UAV data to derive information on the current land use and for monitoring urban developments. Furthermore, the immediateness of the data provision was seen to be very beneficial to investigate and solve land disputes within group ranches. However, since the registry index maps are paper-based, the entry barrier to adopt UAV technology is very high. Good visibility of rooftops and information on the height of buildings was found to support land valuation processes. The exploratory case study in Kenya showed that most of the 10 aspects can be met. As an indirect surveying technique, the concept of using UAV technology in cadastral mapping is based on the existence of visible boundaries which can either be extracted by automated image analysis or manual delineation. However, it was also found that a precise and accurate generated orthophoto allows

extraction of boundaries that are not necessarily visible, such as combining features that demarcate the corner point of the parcel even though the line in between is not visible. The ease of use and the flexible setup in terms of the technical standards of the sensor and platform allows covering a large range of different purposes. In terms of scalability, UAV technology only serves a limited range of different scales as costly and lengthy flight authorization procedures hinder an efficient application. Furthermore, in many countries current regulations require to fly missions which are in visual line of sight, allowing only some hundred meters of a possible flight trajectory. According to Kenyan stakeholders, limited battery capacity was found to be the second bottleneck currently impeding large scale implementation.

For ABE, the results from the workshop proved that our proposed automatic boundary extraction approach facilitating the delineation of visible objects and cadastral boundaries can be used to collect information on land tenure, land value, and land use. It further aligns well with the FFP spatial and scalability requirements: it allows a cheap, fast, and accurate delineation of visible boundaries from aerial imagery. However, costs, speed, and accuracy can vary depending on the capture and processing of the aerial imagery and the implementation of the automated boundary extraction: the approach is currently open source, which seems low-cost, but might require more time in acceptance as the SWOT analysis revealed. Given the complexity of cadastral boundaries, automating their delineation remains challenging: the variability of objects and extraction methods reflect the problem's complexity, consisting of extracting different objects with varying characteristics. These circumstances impede the compilation of a generic model for a cadastral boundary and thus the development of a generic method. These remarks come back to the limitations of general boundaries: no standardized specifications exist for boundary features, boundaries are often not marked continuously and maintained poorly [59]. To further develop automated boundary extraction in indirect surveying, we suggest considering the extractable boundary rather than the visible boundary alone (Figure 14): instead of focusing on the visible boundary comprising of outlines of physical objects, automated boundary extraction should focus on the extractable boundary that incorporates local knowledge and context. This information is not inherent in the concept of the visible boundary, but it is extractable from remote sensing imagery.

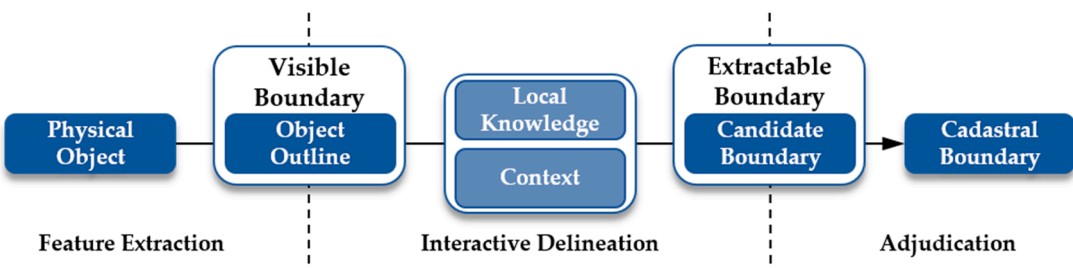

**Figure 14.** From physical object to cadastral boundary: reformulated boundary concepts for indirect surveying.

Overall, our approach that couples a machine-based automatic feature extraction with a delineator-based interactive delineation can be used to map extractable boundaries. The delineation cannot be fully automated at the current state since the extracted outlines require (legal) adjudication and incorporation of local knowledge from human operators to create final cadastral boundaries. Image-based approaches bear potential to automatically extract use rights, which do not necessarily represent legal rights. These circumstances limit the scope of automated approaches. We observed that automating boundary extraction dealing with sensitive land rights can only be successful, when the interactive part that bridges the gap between automatically generated results and the final cadastral boundary is designed and implemented in correspondence to user needs. Our work revealed limitations of the current approach and ideas for improvements to be addressed in future work, in order to advance the current approach regarding efficiency and acceptance. This would promote the paradigm shift towards cadastral intelligence that integrates human-based expert knowledge with automatically

generated machine-based knowledge. Additionally, future studies should provide approaches to capture requirements from existing technical, legal, financial, and institutional frameworks to be considered when aiming to implement innovative cadastral mapping procedures successfully.

Finally, for governance aspects, further notes on legislative and financial aspects are worth expanding upon. Implementing any of the remote sensing methodologies at any scale, without an appropriate legislative framework, appears fraught. This partly relates to modernizing existing laws and regulations to open up to innovative approaches, and partly to new rules for new challenges. Especially when this new legislation would give clarity on the responsibilities of the different actors, prioritize cheap and open source technologies and stimulate and facilitate partnerships between the governmental and non-governmental actors that would make the uptake and upscaling of the remote sensing methodologies much more likely. Without this an occasional "pilot" might continue and show what can and cannot be achieved within a certain setting, but for true upscaling, a supportive environment will be needed; appropriate laws and regulations and a collaborative attitude among national and local government, as well as with non-government actors. As Kenya has a long land administration history, there is the human capacity in the field, however it seems this country is lacking the political will to introduce that supportive environment to a large extent. Focusing specifically on UAV legislation, getting to balanced legislation that allows a responsible use of UAVs without truly compromising the other issues is not easy. This can be seen worldwide, but even more in countries like Kenya, which struggle to get political will to make clear instructions for UAVs. The implementation of the UAVs could be improved by increasing collaboration between the national and local governments with the non-governmental actors. This collaboration could help to solve the lack of important financial resources. Resources are needed to hire new staff, training, certification, among others. In this sense, the national level can also play an important role as a facilitator to allow private companies to participate. Some non-governmental actors such as private companies could have the resources to use the UAVs; however, they require certain governmental support such as the issuing of the permits or incentives to invest. Meanwhile, on the financial aspect, both proprietary and open source options present challenges: actors payments for software, licenses, and the required updates prohibitive; however, even with open source software, the lack of IT infrastructure and internet access still impacts negatively on scaled uptake. In Kenya, the current resources are not enough to establish a sustainable implementation.

## 6. Conclusions

The paper has described challenges around land tenure mapping in Kenya and presented potential remote sensing methodologies that respond to current end user needs and that are furthermore investigated from a governance perspective. Although the 2010 Constitution resulted in a land policy reform setting out a framework to better respond to the needs of large customary groups in Kenya, actual implementation is slow and both county governments and communities themselves continue to grapple with a multitude of issues relating to rapid urbanization, unmanaged development, and unregulated land activities. Communities are not engaged with land policies, and spatial planning and needs are not being met. For counties like Kajiado, these challenges are further exacerbated by issues of scale, high levels of corruption and poor-quality of existing land data. In future, since the needs are changing with time the new technologies in support of land administration should definitely be adapted accordingly.

With regards to the demonstrated remote sensing methodologies, SmartSkeMa was revealed as a versatile land data acquisition tool that requires little expertise to be used and is based on community participation; UAVs were identified as having a high potential for creating up-to-date base maps to support the current land administration system; the automatic boundary extraction approaches designed for areas demarcated by physical objects and are thus visible were found to be useful for collecting information on land tenure, land value, as well as land use (aligned with the 10 aspects).

Finally, with regards to ensuring responsible governance related to the scaled implementation of the remote sensing methodologies, as there is no appropriate legal framework for applying them,

the viability and medium timeframe for increased usage in the sector remains unclear. A more robust legal framework could strengthen authority; operationalize in administrative orders, rules and planning; and serve as the basic control system (for possible sanctions). After establishing the framework, if not during, serious attention needs to be given to the cooperation between all relevant actors, where interorganizational relations are ruled by the acknowledgement of mutual interdependencies, trust and the responsibilities of each actor.

**Author Contributions:** Conceptualization: M.K., C.S., S.C., S.H., M.C., J.S., R.B., J.Z., C.L., and J.C.; Funding acquisition, R.B.; Visualization: M.K., C.S., S.C., S.H., M.C., and J.S.; Writing—original draft: M.K., C.S., S.C., S.H., M.C., J.S., and I.B., Review & editing: M.K., C.S., S.C., S.H., M.C., J.S., R.B., J.Z., G.V., C.L., J.C., I.B., G.W., and V.P., Fieldwork support: G.W., R.W., P.O.O., G.T.O., and B.C. All authors have read and agreed to the published version of the manuscript.

**Funding:** The research described in this paper was funded by the research project "its4land", which is part of the Horizon 2020 program of the European Union, project number 687828.

**Acknowledgments:** Its4land team would like to express also their acknowledgements to Gordon Wayumba, Robert Wayumba, Peter Ochieng Odwe, George Ted Osewe, Beatrice Chika, and their colleagues for their support during the fieldworks and workshops.

**Conflicts of Interest:** The authors declare no conflicts of interest.

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
