# Peer review of "Innovative Remote Sensing Methodologies for Kenyan Land Tenure Mapping"

_remotesensing, doi:10.3390/rs12020273_

Round 1
Reviewer 1 Report
The article could have been slightly shorter.
Author Response
We would like to express our gratitude to the reviewer for the positive evaluation of our work and for the time spent and efforts to provide us with helpful comments. We have addressed all indicated suggestions and recommendations. We will provide a new version of the paper with the changes according to reviewers' suggestions.
Please note that the line numbers in this response correspond to the file which is showing the track changes (all markup). We will provide a file without track changes where the figures and empty spaces are arranged.
Comment:
The article could have been slightly shorter
Response:
The paper has been shortened and slightly changed based on the comments from the reviewers (lines 40 to 50; 232 to 234; 166 to 168; 582 to 585 – file version with track changes).
Reviewer 2 Report
This paper discusses the application of remote sensing methodologies to support land rights mapping. It assesses user needs (in terms of land administration functions), and how the remote sensing methodologies can meet these needs. It also considers how the adoption of these technologies may have governance implications.
The paper is in a good structure and very well written.
General feedback:
Abstract is very long. Remove lines 40-50.
The relation of this paper to its4land project should be mentioned in Acknowledgements.
Line 74- The reference used for the fact ‘only 30% of the world’s population has documented land rights’ goes back to 2013. The authors are recommended to validate this fact and use a more recent reference.
Line 137 – merge the two references like [29, 30] instead of [29] [30].
What do the colours show in Fig 1? There’s a need for a legend. The image quality is very poor.
Figure 2 – why isn’t ‘land development’ considered under LA functions?
Line 200 – The last sentence starting with ‘Needs’ is incomplete.
Line 201 – change section 3.2 title to ‘Remote sensing methodologies’
Line 208 – change ‘contexts’ to ‘context’
Line 211 mentions that the procedure for testing each methodology is shown in Figure3 below. However, the figure only shows the pictures of the workshop, not the procedure.
Figure 5 – LADM is mentioned in the caption of this figure for the first time. Is there any LADM designed for Kenya? If so, it needs to be discussed earlier in the paper.
Figure 7 quality is very poor.
Line 323 - As FFP is used before, no need to spell it out again (Fit-for-purpose)
Line 334- replace ‘were’ with ‘was’
Table 1- add ‘needs’ after ‘County Government’ and ‘Communities’ column names
Line 452- what’s the starting word ‘wever’? Is it ‘however’?
Line 479, (h) needs to be mentioned in the text too.
Line 504, replace ‘click’ with ‘clicks’
Line 508, replace ‘reduces’ to ‘reduced’
Reword lines 559-561 sentence.
Line 602, replace ‘it’s’ with ‘its’
Line 645, replace ‘reflects’ with ‘reflect’
Figure 11 quality is very poor.
Line 713, reword the sentence containing ‘could be support’
Future direction of the research needs to be added to Conclusions.
Author Response
We would like to express our gratitude to the reviewer for the positive evaluation of our work and for the time spent and efforts to provide us with helpful comments. We have addressed all indicated suggestions and recommendations. We will provide a new version of the paper with the changes according to reviewers' suggestions.
Please note that the line numbers in this response correspond to the file which is showing the track changes (all markup). We will provide a file without track changes where the figures and empty spaces are arranged.
The paper is in a good structure and very well written.
General feedback:
Abstract is very long. Remove lines 40-50. – Done (lines 40 to 50 – file version with track changes) The relation of this paper to its4land project should be mentioned in Acknowledgements. – Done Line 74- The reference used for the fact ‘only 30% of the world’s population has documented land rights’ goes back to 2013. The authors are recommended to validate this fact and use a more recent reference. – Done (Additional reference from 2016 was added) Line 137 – merge the two references like [29, 30] instead of [29] [30]. - Done What do the colours show in Fig 1? There’s a need for a legend. The image quality is very poor. – Done (The image quality was improved and legend was added – line 190) Figure 2 – why isn’t ‘land development’ considered under LA functions? - These four functions (line 197- 200) are quite interrelated. The fourth function “land development” is representing a process of changes over time therefore it is not within the focus of the current research Line 200 – The last sentence starting with ‘Needs’ is incomplete. - Done Line 201 – change section 3.2 title to ‘Remote sensing methodologies’ - Done Line 208 – change ‘contexts’ to ‘context’ - Done Line 211 mentions that the procedure for testing each methodology is shown in Figure3 below. However, the figure only shows the pictures of the workshop, not the procedure. – Done (lines 232 to 234 – file version with track changes) Figure 5 – LADM is mentioned in the caption of this figure for the first time. Is there any LADM designed for Kenya? If so, it needs to be discussed earlier in the paper. Done (lines 166 to 168 – file version with track changes) Figure 7 quality is very poor. – Done (The quality of the figure was improved) Line 323 - As FFP is used before, no need to spell it out again (Fit-for-purpose) – Done (line 346) Line 334- replace ‘were’ with ‘was’ - Done (line 357) Table 1- add ‘needs’ after ‘County Government’ and ‘Communities’ column names - Done Line 452- what’s the starting word ‘wever’? Is it ‘however’? – Done (It is However) Line 479, (h) needs to be mentioned in the text too. – (h) is explained on line 498 Line 504, replace ‘click’ with ‘clicks’- Done Line 508, replace ‘reduces’ to ‘reduced’- Done Reword lines 559-561 sentence. – Done (lines 582 to 585 – file version with track changes) Line 602, replace ‘it’s’ with ‘its’ – Done Line 645, replace ‘reflects’ with ‘reflect’ – Done Figure 11 quality is very poor. – Done (The quality has been improved and the number of the Figure has been changes to 13) Line 713, reword the sentence containing ‘could be support’ - Done (lines 741in the file version with track changes) Future direction of the research needs to be added to Conclusions. - Done (lines 735 to 737 – file version with track changes)
Reviewer 3 Report
Suggestions for Authors:
- In the second chapter, you can add a drawing showing the example of current land registration,
- What extent can the instability of law in the field of UAVs cause restrictions in the use of this method?
- Were the results obtained consulted with the owners of individual plots covered by the study?
Author Response
We would like to express our gratitude to the reviewer for the positive evaluation of our work and for the time spent and efforts to provide us with helpful comments. We have addressed all indicated suggestions and recommendations. We will provide new version of the paper with the changes according to reviewers suggestions.
Please note that the line numbers in this response correspond to the file which is showing the track changes (all markup). We will provide a file without track changes where the figures and empty spaces are arranged.
Response to Comments
Comment 1: In the second chapter, you can add a drawing showing the example of current land registration,
In the second chapter (Case Background and Study Area) more information of the current registration system is added (see lines 150 -152 file version with track changes) with a reference showing a table with the organization structure. In addition lines 162-168 (file version with track changes) are added explaining the types of maps that the current cadastre contains and pictures from the archive obtained during the fieldwork from the land professionals are added.
Comment 2: What extent can the instability of law in the field of UAVs cause restrictions in the use of this method?
The instability in policy and legal environments definitely impacts directly on the ability to utilize UAVs and undertake flights. However, because the use of UAVs has been illustrated recently, for many applications, many countries are establishing regulations for their use. In Kenya, regulations were gazetted in October 2017 but were nullified in 2018 as it was argued that the regulations did not serve the society: the financial and operational barriers were considered too high. This could be considered quite an extraordinary situation, however, it at least appears to be a good explanation and motivation that the government is seeking to support the democratic and more wide spread use of UAVs. At any rate, with regards to the work at hand, with support colleagues TUK, special permission from the national government and the county government were obtained and data collection and test flights could be undertaken. Data collection and dissemination workshops were held in September/October 2018 in Kajiado and Nairobi to evaluate the overall performance of the UAV-based workflow. Current regulatory barriers aside, from the results and feedback, obtained, it is believed that regulations will be gazetted in the short to medium term: the evidence for UAV application and utility is increasingly overwhelming.
Comment 3: Were the results obtained consulted with the owners of individual plots covered by the study?
The majority of all administrative leaders and the owners were consulted prior to the data acquisition, to obtain permission for the flight. Owners were also consulted to get an understanding of local boundary characteristics. These stakeholders were able to show examples of visible boundaries. Images were made images of different boundary types and this provided insights about different local demarcation challenges. After the automatic boundary delineation, it is important to note that it was impressed upon the stakeholders that we were not creating legal boundaries. The procedure was merely demonstrating, for the workshop to the mapping specialists, innovative methods to extract visible boundaries in an automatic way, based on VHR UAV images. Therefore, a final confirmation and consultation session with owners were not formally organized, as it was outside the scope of the study.
